# Fluorescence Approaches for Characterizing Ion Channels in Synthetic Bilayers

**DOI:** 10.3390/membranes11110857

**Published:** 2021-11-04

**Authors:** Md. Sirajul Islam, James P. Gaston, Matthew A. B. Baker

**Affiliations:** 1School of Biotechnology and Biomolecular Sciences, University of New South Wales, Kensington, NSW 2052, Australia; md_sirajul.islam@unsw.edu.au (M.S.I.); james.gaston@student.unsw.edu.au (J.P.G.); 2CSIRO Synthetic Biology Future Science Platform, GPO Box 2583, Brisbane, QLD 4001, Australia

**Keywords:** ion channel, liposome, synthetic bilayer, membrane protein, rhodopsins, reconstitution, fluorescence assay, *p*H-sensitive dyes

## Abstract

Ion channels are membrane proteins that play important roles in a wide range of fundamental cellular processes. Studying membrane proteins at a molecular level becomes challenging in complex cellular environments. Instead, many studies focus on the isolation and reconstitution of the membrane proteins into model lipid membranes. Such simpler, in vitro, systems offer the advantage of control over the membrane and protein composition and the lipid environment. Rhodopsin and rhodopsin-like ion channels are widely studied due to their light-interacting properties and are a natural candidate for investigation with fluorescence methods. Here we review techniques for synthesizing liposomes and for reconstituting membrane proteins into lipid bilayers. We then summarize fluorescence assays which can be used to verify the functionality of reconstituted membrane proteins in synthetic liposomes.

## 1. Introduction

Ion channels provide a pathway for the movement of ions into and out of cells and organelles in all living organisms [1]. They are involved in many important cellular processes including membrane shaping and stabilization, immune response, and muscle contraction [1,2,3]. They are also critical to all cellular communication in nerves, muscles, and synapses [4,5,6]. Understanding the functioning of ion channels is of the utmost importance not only due to their role in essential cellular processes [1,2,3,4,5,6], but also for their importance as molecular targets for drug development [7,8,9].

Detailed study of the ion channels is frequently challenging due to the complexity of the membrane in which they are embedded [10,11,12]. In recent years, in vitro studies have been used to disclose various features of reconstituted ion channels in synthetic lipid bilayers [13,14]. Moreover, in vitro assays offer control over the entire composition of the system, which is difficult within in vivo assays. To investigate how membrane-protein ion channels are affected by the surrounding lipid environment, isolation and reconstitution of these proteins into model lipid membranes is required [15,16]. Different systems for reconstitution allow for various biochemical and biophysical techniques for characterizing the ion channels [17].

Of particular interest are techniques that leverage fluorescence microscopy to measure spatiotemporal phenomena in high throughput at high resolution [18]. The flow of the ions through the lipid membranes of vesicles can be quantified by fluorescence simultaneously while fluorescent dyes are used as indicators to examine the activity of the reconstituted proteins [19,20].

Here we give an overview of the types of techniques for synthesizing liposomes. We also summarize recent approaches for membrane protein reconstitution into lipid bilayers and the limitations of each method. Finally, we discuss fluorescence assays which can be used to verify the functionality of the reconstituted membrane proteins and the corresponding strengths and weaknesses of such assays.

## 2. Lipid and Membrane Bilayers

Membrane bilayers form the basis of compartmentalisation within biological cells and systems [21]. Of particular interest are the cellular membranes of living organisms, which are composed of a combination of lipids and membrane proteins. Lipids are a diverse range of molecules that are generally hydrophobic or amphipathic in nature [22]. Lipid structure can be divided into polar hydrophilic ‘head’ groups which are composed of phosphates (phospholipids) or carbohydrates (glycolipids), and hydrophobic ‘tail’ groups which are made up of acyl chains that are either saturated or unsaturated [23]. Within aqueous environments, lipids have the ability to form spherical bilayers, with their head group facing outwards and their tails facing inwards. This results in the formation of a hydrophobic environment encased withing the lipid membrane leaflet. Bilayer formation varies based on the types of lipids used, along with how the lipid compositions interact with each other. For example, positively charged amphiphilic lipids are poorly soluble in water, so they often need to be dissolved in other solvents such as chloroform and driven into forming membrane structures [24]. One membrane bilayer structure that is of particular interest is liposomes. Liposomes are vesicles formed by an aqueous interior and a surrounding membrane bilayer, usually formed of amphiphilic phospholipids and cholesterol (Figure 1B) [25]. Important areas in liposome research include methods for producing liposomes of varying sizes and structures, applications of liposomes such as encapsulation of molecules within the aqueous interior, protection and targeted delivery of therapeutics [26,27], and lastly the use of liposomes as synthetic models for studying the interactions of cellular membranes and membrane proteins.

### 2.1. Liposomes

Liposomes encompass a broad range of bilayer structures that can be formed from natural and synthetic lipids, with their classification varying based on size and structure. The most widely used are unilamellar liposomes, which are composed of a single membrane bilayer, and can be further classified based upon their size. Classifications include small unilamellar vesicles (SUVs, <100 nm), large unilamellar vesicles (LUVs, 100–500 nm), or giant unilamellar vesicles (GUVs, ≥1 µm) (Figure 1A) [23]. Other types of liposomes outside of unilamellar include multilamellar vesicles (MLVs) which are formed of multiple concentric bilayers, and multivesicular liposomes (MVLs) which are formed of multiple non-concentric bilayers or interconnected monolayers within a larger membrane vesicle (Figure 1C,D) [28].

### 2.2. Synthesis of Liposomes

There are various methods for producing each type of liposome. MLVs can be formed by dissolving lipids in an organic solvent such as chloroform, and using a nitrogen stream to subsequently create a lipid film [29]. They can also be formed through evaporating solvent emulsion droplets containing lipids from an agitated aqueous mix [30], or from the addition of proliposomes (liposomes encapsulated in a salt or sugar granule) to aqueous solution [31]. Both SUVs and LUVs can be formed using multiple methods such as extrusion of a lipid mixture through a polycarbonate filter, detergent dialysis (detergent-stabilized lipids form liposomes as detergent is removed), sonication of MLVs into smaller liposomes, ethanol injection (ethanol is rapidly injected into lipid-buffer solution), and freeze-thawing to form LUVs [25]. GUVs can be formed using electroformation, where lipids form GUVs on an electrode surface by inducing an electro-osmotic effect from a voltage current [32]. They can also be produced by swelling hydrogel lipid films [33], gentle hydration of dried lipid films [34], and emulsion-based methods [35].

### 2.3. Microscopy

The method of microscopy used for visualisation of liposomes is dependent on the liposome size and type. Both traditional light microscopy and phase-contrast microscopy can be used to visualize GUVs and larger liposomes [36], but these methods are unable to provide details beyond broad morphology and heterogeneity [37]. Electron microscopy has been used for imaging all sizes of liposomes, allowing for characterization of bilayer structures in liposomes as small as SUVs [37]. This powerful visualisation of bilayers does have its drawbacks, however. Negative-staining transmission electron microscopy (TEM) involves particles being adsorbed to a carbon film grid, surrounded by a heavy metal salt, and quickly air-dried during which the heavy metal salt forms an amorphous film embedding the particles of interest in the process [38]. This process is susceptible to causing changes to the original vesicular structure and the stains themselves can lead to the formation of dark and light fringe artifacts which could be misinterpreted as lamellar structures [39]. Cryo-electron microscopy is an alternative that does not require staining staining, instead using a thin film of liposome suspension (usually < 500 nm) that is plunge-frozen to allow for fast freezing, creating a thin layer for imaging [38]. During this process rearrangement and flattening of larger liposomes is possible [38], with typically fewer liposomes able to be imaged than via TEM [40].

Fluorescence microscopy provides a means for characterizing liposomes of all sizes and morphologies. SUVs, LUVs, and GUVs have the capacity to be fluorescently labelled in various ways to visualize liposomes with minimal disruption to membrane stability [41]. The use of fluorescent dyes, such as lipophilic carbocyanine dyes, allows for the intercalation of fluorescent labels directly into the membrane bilayer [42]. The head-groups of lipids can also be directly labelled with fluorescent probes, which can then be incorporated as part of the membrane bilayer [43]. These modes of fluorescent labelling are suited for high-throughput and single-molecule super-resolution microscopy methods, such as photoactivation localisation microscopy (PALM) [44] and total internal reflection fluorescence (TIRF) microscopy [45]. For GUVs, encapsulated fluorescent proteins like GFP can be used as a means of visualising the internal liposome environment [46]. Their larger size also makes them suitable for confocal microscopy as a means of eliminating background noise, controlling the focal plane, and observing 3D z-stack images to better characterize labelled membrane structures [47]. Fluorescence microscopy also has the advantage of being able to observe multiple fluorescently labelled liposomes at once, provided they are sufficiently separated in emission spectra [48]. Furthermore, the implementation of environment-sensitive and self-quenching fluorophores is useful for creating responsive liposome assays that can characterize changing internal and external environments such as the flow of ions across liposome membranes [48,49,50,51,52]. There are some inherent drawbacks of fluorescent labelling such as the requirement for sample modification and potential photobleaching over time [53,54]. However, with the use of appropriate labels, fluorescence microscopy provides a unique opportunity to visualize and characterize dynamically changing bilayers and internally encapsulated environments of a range of liposomes in real time.

## 3. Ion Channels

Ion channels are naturally occuring membrane proteins that allow ions to pass through a channel pore. Ion channels are responsible for the regulation of major physiological functions and can transport hundreds of ions in a second without the help of any metabolic energy. Ion channels are distinct from simple aqueous pores due to their two important properties—they are gated and they show ion selectivity, permitting some ions to pass but not others. Based on ion selectivity, ion channels allow the flow of specific inorganic ions, mainly Na^+^, K^+^, Ca^2+^, or Cl^−^ and are named after these ions [55]. Gating indicates that they have ability to open and close in response to a specific stimulus [56]. Ion channels can be further classified depending on the primary stimulus for opening, e.g., voltage-gated channels, ligand-gated channels, or mechanically gated channels (schematically illustrated in Figure 2). The voltage-gated channels are ion-selective channels that respond to perturbations in cell membrane potential [57] (Figure 2A). Depending on the position of the ligands, the ligand-gated channels are called extracellular ligand-gated or intracellular ligand-gated channels, which are sensitive to specific ligands [58] (Figure 2B). As a result, ligand-gated channels are targets for many drugs including anesthetics [59], antipsychotics, and antidepressants [60]. On the other hand, the mechanically gated channels sense the forces on the cell membrane and transduce the external mechanical forces into electrical and/or chemical intracellular signals [61] (Figure 2C). Mechanosensing is an important target not only for diagnosis but also for therapeutics in cancer progression and arthritis [62,63].

Rhodopsins are widely studied membrane proteins as ion channels because the activity of the rhodopsin-based ion channels can be controlled precisely by light [64,65]. They are seven-transmembrane helix proteins that have an integral membrane protein, opsin, and a chromophore, 11-*cis*-retinal [66,67,68]. Major types of rhodopsins studied as ion channels include bacteriorhodopsin (bR) [69,70], proteorhodopsin (pR) [71], xanthorhodopsin (xR) [72,73], archaerhodopsin (AR) [74], bovine rhodopsin [75], channel-rhodopsin (ChR) [76,77], halorhodopsin (hR) [78], and phototaxis receptor sensory rhodopsin [79]. The bR, pR, xR, and bovine rhodopsins all function as proton pumps, while ChR pumps protons along with other non-specific cations such as Na^+^, K^+,^ and Ca^2+^ [80]. hR is a chloride pump, and sensory rhodopsins use a transducer, mostly another membrane protein, or soluble protein to send signal inside the cell [78,79].

## 4. Study of Ion Channels in Synthetic Liposomes

Due to the complicated cellular environment, the study of membrane proteins is hampered even in the simplest cells. In vitro study is an alternative to reconstitute individual proteins in a defined synthetic liposome [13]. The isolated membrane proteins are typically stabilized in solution by amphiphilic detergents and then reconstituted into a model lipid membrane. Many model membranes such as monolayers, bilayers, liposomes, and nanodiscs have been developed, which closely resemble their biomimetic environments [81]. In the following, we will discuss some commonly used methods of protein reconstitution in model membranes.

### 4.1. Integrating Membrane Proteins into Liposomes

#### 4.1.1. Direct Reconstitution

In the direct reconstitution method, the isolated membrane protein solubilized by detergents is added to preformed liposomes. The detergent facilitates the insertion of proteins in the lipid bilayers by a modest destabilization of the lipid vesicles [82,83] (Figure 3A, method 1). Next, the detergent is removed using insoluble hydrophobic resins, e.g., Sephadex G50 or Bio-Beads [84], resulting in the formation of stable bilayer vesicles with incorporated membrane proteins. This reconstitution method is a fast and frequently used strategy for proteoliposome preparation. However, due to incompatible detergent selection, it can sometimes destabilize membrane protein and result in a loss of function.

#### 4.1.2. Dehydration–Rehydration

This procedure is based on the deposition of a solution of membrane protein and lipids on a solid surface being dehydrated followed by a controlled rehydration (Figure 3A, method 2). Upon rehydration, the formation of the vesicles can be facilitated by spontaneous swelling [85] or applying an electric field (electroformation) [32]. In this method, the proteins reconstituted in the giant liposomes are from one small liposome with already-reconstituted proteins [86]. In the electroformation process, the lipid mixture is generally deposited on glass slides coated with indium tin oxide or platinum wires. This method of the preparation of giant liposomes and also proteoliposomes is simple and can be performed under physiochemical conditions. The dehydration step should be controlled otherwise the protein can be denatured.

#### 4.1.3. Induced Fusion

In this method two liposomes, usually one with already-reconstituted protein, are fused by different inducing factors. A peptide is commonly anchored on the liposomal membrane to induce the fusion [87] (Figure 3A, method 3). The fusion can also happen spontaneously, but requires specific conditions (Figure 3A, method 4). More recently, complementary lipidated DNA has been employed for DNA-programmable controlled fusion of liposomes. Induced fusion is very fast and can be controlled precisely. However, conditions should be optimized, otherwise the peptide or fusing agent can be embedded in the lipid bilayer, which interrupts the study of the transport across ion channels.

#### 4.1.4. Microfluidic Jetting

In this technique, the vesicles with reconstituted membrane proteins are formed in three stages: (I) initial membrane protrusion in a chamber where two planar monolayers are formed separated by a spacer, (II) membrane collapse by removing the spacer and encapsulation, and (III) separation of the vesicles from the membrane using a microfluidic jet [88,89] (Figure 3A, method 4). This is a technique used to produce liposomes with controlled size and membrane composition, membrane protein incorporation, and encapsulation. The use of this method is due to the requirement for costly, specialized equipment.

### 4.2. Integrating Membrane Proteins into Nanodiscs

Nanodiscs are another frequently used model system for the reconstitution of membrane proteins. They are formed spontaneously upon detergent removal from a mixture of membrane scaffold proteins that are derived from apolipoprotein A1 and detergent-solubilized lipids [90,91]. Detergent removal is a critical step to initiate the formation of nanodiscs and this can be accomplished using detergent-absorbing columns or beads or dialysis [84]. In the nanodiscs, the small patch of lipid bilayer is surrounded by two copies of membrane scaffold protein. Membrane proteins are included in the mixture for reconstitution into the nanodiscs (Figure 3B). This method of protein reconstitution is advantageous as the proteins are accessible from both sides which can help when studying ligand-binding interactions or binding of signaling molecules. One limitation of nanodiscs is their small size. The sizes of a nanodisc can be controlled by modifying the lengths of the membrane scaffold proteins in a range of 10 to 20 nm [92], however this is sometimes too small for large membrane proteins or protein–protein complexes. Recently, to overcome the size limitation, peptide-based nanodiscs and polymer-based lipid nanodiscs have been reported, where membrane scaffold proteins are replaced by the peptides or polymers [93,94,95].

### 4.3. Integrating Membrane Proteins in Cell-Free Systems

In a cell-free system, membrane proteins are synthesized in an in vitro system that includes protein translation machinery [96,97]. Several cell-free systems based on *Escherichia coli* cell-extracts with minimal components for synthesis, e.g., the “protein synthesis using recombinant elements (PURE)” system have been reported [98]. In this system, the membrane proteins are translated in the presence of a different biomimetic model lipid system, into which the synthesized membrane proteins are reconstituted. Translation of membrane proteins directly in the presence of liposomes is an attractive approach used for reconstitution for several membrane proteins that include bacteriorhodopsin, connexin, and stearoyl-CoA desaturase [99,100,101]. This method provides co-translational reconstitution into lipid bilayers without the need for detergents.

### 4.4. Integrating Membrane Proteins with Detergent Alternatives

Although detergent is commonly used in the isolation and reconstitution of membrane proteins, recently, amphipathic polymers (amphipols) have been emerged as a detergent-free approach to stabilize and reconstitute membrane proteins [102]. Amphipols consist of many hydrophilic and hydrophobic groups that likely form toroids around the transmembrane domains of membrane proteins with their nonpolar groups, while the outer polymer surfaces comprising polar groups keep them water-soluble [103]. PMAL-B-100 and A8-35 are the most extensively studied amphipols, and have been used to solubilize and stabilize a range of proteins including mitochondrial supercomplex and diacyl-glycerol kinase [104,105]. A copolymer prepared from a 3:1 molar ratio of styrene to maleic acid (3:1 SMA) has also been reported to stabilize lipid particles (termed Lipodisq) and shown to extract and reconstitute bR, hR, and PagP for their functional studies [106,107,108,109].

## 5. Techniques to Characterize Ion Channels in Liposomes

### 5.1. Fluorescence Assays for Characterizing Ion Channels

Directional ion flow through a channel across a liposome membrane is the best validation for correct insertion and function. Considering rhodoposin as a case study which has been extensively reviewed elsewhere [110], there is a controlled sequence of events involving alternating light and dark cycles that are used to determine if this ion channel has been successfully and functionally inserted (Figure 4A). Upon illumination with light of the correct wavelength (550 nm for bR), rhodopsin will become activated and will change the retinal conformation from all-*trans* to 11-*cis*. This conformational change opens the channel and allows protons to flow through the channel. When the light source is withdrawn (the sample is placed in the dark), protons will diffuse across the membrane until the external and internal *p*H of the liposome equilibrates. When illuminated once again, rhodopsin activates, and proton flow recommences. The flow of the protons can be monitored by *p*H-sensitive fluorescent dyes as a function of time, and we describe three broad categories of *p*H-sensitive dyes: (1) hydrophobic membrane-permeable fluorescent *p*H-indicators, that can be added in the bulk solution of proteoliposomes, (2) water-soluble, membrane-impermeable *p*H-sensitive dyes, that can be encapsulated in proteoliposomes, and (3) lipids with a *p*H-sensitive head group, that can be included in the lipid mixture (Figure 4B). Different *p*H-sensitive dyes commonly used to monitor proton pumping assays are summarised in Table 1.

#### 5.1.1. Membrane-Permeable Dye-Based Assays

Membrane-permeable *p*H-sensitive dyes are typically hydrophobic. Some of these include acridine orange and ACMA (9-amino-6-chloro-2-methoxyacridine). These dyes are added to the bulk solution of the sample. Due to their membrane permeability, they diffuse through the membrane of the liposomes and an equilibrium is established across the bilayer. Upon protonation inside the liposome, their fluorescence is quenched, and they become membrane impermeable. ACMA has been used in several studies to validate the functionality of different rhodopsins. Hoi et al. reported an ACMA-based fluorescence assay to verify the proton-pumping activity of the reconstituted mCitrine-tagged ChIEF, an engineered variant of ChR, into LUVs [111] (Figure 4C). In their assay, K^+^ ionophore valinomycin was added to allow leakage of K^+^ from the interior of the liposome, at high concentrations, to the outside. It enabled the exchange of membrane potential (∆ψ) for ∆*p*H due to H^+^ loading inside the liposome. Following illumination, protons were then conducted through the open channel to balance the inside-negative membrane potential. As a consequence of increased proton concentration in the interior, the fluorescence of ACMA decreased inside the vesicle. Finally, the protonophore CCCP was added as a control to allow free diffusion of protons across the membrane, resulting in a dramatic fluorescence drop (Figure 4D). Using ACMA, a similar assay was used to study the proton transport activity of voltage-dependent H^+^ (Hv1) channels, vacuolar class of (H^+^)-ATPases (V-ATPases), and F_O_F_1_ ATPase [112,113,114,115]. Recently, an artificial photosynthetic cell was reported in which ATP synthase and bR were used as an energy-generating system [116]. The proton-pump activity of the reconstituted bR was monitored by the fluorescence change of ACMA. The functional activity of the TREK-1 and TWIK-1 (potassium ion channel) and CLC (chloride channel) were also monitored by ACMA, where the protonophore CCCP was added to initiate the flux of potassium [117,118,119]. ACMA is suitable for relative measures of *p*H, but it cannot typically be used to quantify protonation as the total amount of ACMA inside the membrane may change as new dye equilibrates across the membrane.

#### 5.1.2. Membrane-Impermeable Dye-Based Assays

Membrane-impermeable dyes like pyranine (8-hydroxypyrene-1,3,6-trisulfonic acid, HPTS) or SNARF (seminaphtho-rhodafluor) are water soluble. These are typically encapsulated inside the proteoliposomes [120,121]. Due to their hydrophilic properties, these dyes do not redistribute upon protonation and thus are convenient for measurement and comparisons between protonation in the interior and exterior of a liposome [122,123]. The experimental setup is otherwise similar to that shown in Figure 4C for ACMA. Pyranine has been used as a *p*H-sensitive dye in much rhodopsin work [124]. Verchere et al. reported a protein activity assay, where the MexAB protein transporter and bR were reconstituted into liposomes [125,126] (Figure 4E). The MexAB was energized and activated by the proton gradient generated by bR. Using pyranine, the internal *p*H change of the liposome was measured as a function of time (Figure 4E). Pyranine was also used to confirm the activity of bR reconstituted in LUVs, GUVs, large proteoliposomes, and ABA block copolymer vesicles [83,86,127,128]. Pyranine has been used to monitor the proton-pumping activity of different types of rhodopsins: sensory rhodopsin-I and II [129,130], xR [131], ChR [132], aR [133], pR [134], and xanthorhodopsin [135,136]. Recently, Ghilarov et al. revealed the molecular mechanism of the membrane protein SbmA [137]. The proton-pumping activity of the protein reconstituted into liposomes was measured by pyranine fluorescence. In another pyranine-based assay, a synthetic phospholipid LC ion channel for Rb^+^ has been investigated using FCCP protonophore [138]. One issue with pyranine assays is the low dye entrapment efficiency in detergent-based reconstitution methods. To increase the efficiency, the dye concentration is kept high during the whole reconstitution procedure. This then requires a further step of gel filtration or dialysis to separate the external dye from the proteoliposomes.

#### 5.1.3. Lipid-Coupled Dye-Based Assays

Dyes that are covalently linked to individual membrane lipids can be used to measure *p*H locally near liposomes. In these experiments, dye-coupled lipids are mixed with other lipids prior to liposome formation. This process requires no extra effort for dye encapsulation and also no redistribution of dye that can complicate the measurement of *p*H in other approaches. Fluorescein, Oregon Green 488, and *p*Hrodo are examples of *p*H-sensitive dyes that have been used to measure *p*H. Bolli and co-workers synthesized fluorescein-coupled phosphatidyl-ethanolamine (fluorescein-PE) and used it to monitor the proton-pumping activity of cytochrome c oxidase [140]. Kemmer et al. synthesized the *p*Hrodo-DOPE from *p*Hrodo succinimidyl ester and 1,2-dioleoyl-sn-glycero-3-phosphoethanolamine (DOPE) [139] (Figure 4F). The *p*Hrodo-DOPE was used to observe the *p*H change of the interior of the vesicles as a result of the proton-pumping activity of the plant H+-ATPase and P-type ATPase AHA2 [139,141]. Oregon Green 488 was also coupled to a lipid (1,2-dihexadecanoyl-sn-glycero-3-phosphoethanolamine, DHPE) and is commercially available. Using Oregon Green 488 DHPE, the proton-pumping activity of the reconstituted F-type ATPase was observed by monitoring the interior *p*H changes of the proteoliposome as a function of time (Figure 4E) [142]. Oregon Green 488 DHPE was also used to quantify the Hv1-induced proton transport [143].

### 5.2. Non-Fluorescent Assays for Characterizing Rhodopsins

Label-free assays overcome some inherent drawbacks of fluorescent labelling, such as the requirement for sample modification and photobleaching. The commonly used non-fluorescent techniques are electrochemical analysis, surface plasmon resonance (SPR), and plasmon-waveguide resonance spectroscopy. Nekrasova et al. reported a method to synthesize bR in large quantities using *E. coli* [147]. The light-dependent proton flux of the reconstituted bR in lipid vesicle was monitored electrochemically. The change of the *p*H of the solution containing proteoliposome with bR was monitored with continuous stirring by a glass electrode connected to a *p*H meter. In another approach, pR was engineered with red (mCherry-pR) or green (GFP-pR) fluorescent proteins for the guided insertion of the pR into liposome [148]. The photoactivity of the engineered pR was monitored electrochemically in a temperature-controlled system where the *p*H was measured using a micro *p*H-electrode. Rhodopsin was also reported to reconstitute into the supported lipid bilayers (SLB), with the aim of being used as biosensors and biochips. One study showed bovine rhodopsin being successfully reconstituted into a SLB of bis-sorbylphosphatidylcholine [149]. The activity of rhodopsin was monitored by plasmon-waveguide resonance spectroscopy, a powerful tool for studying membrane signalling without the need for labelling of proteins. In an assay of surface-sensitive detection, bovine rhodopsin was immobilized into a SLB. Also, a hydrogel-supported lipid bilayer was used to reconstitute rhodopsin. Both of these systems were characterized by SPR. SPR allows real-time, label-free detection of biomolecular interactions, where the functionality of the rhodopsins is characterized by the direct binding of ligands on the immobilized reconstituted rhodopsins in the lipid bilayer [150,151].

## 6. Limitation of Fluorescent Assays for Characterizing Ion Channels

Fluorescence-based assays allow high throughput screening of ion channels. Moreover, they are easy to implement and to optimize high signal to noise ratio. However, several limitations persist. One major limitation is the low temporal resolution. The change of fluorescence of the dyes occurs upon binding to specific ions. This redistribution process can take minutes and thus, these assays are not suitable to monitor the kinetics of ion channel transitions that can occur on the scale of milliseconds. Fluorescence methods cannot easily resolve opening, activation, inactivation, and sub-steps that occur during these processes. In fluorescence assays, ion-concentration-dependent changes of fluorescence signals are measured, but the ionic current is not directly measured. On the other hand, electrophysiological methods, particularly the patch-clamp, directly measure the ion channel currents in the range of picoamperes over microseconds [152]. Such methods are thus better for assessing conformational changes that occur on these timescales in vivo and in vitro, although these techniques are typically low throughput, require specialized instruments and expertise, and are not suited to liposomes <1µm in diameter [153,154,155]. Further, due to the low sensitivity of fluorescence methods, subtle perturbations on ion channels may not be detectable. Examples include perturbations to the magnitude of ionic currents in K^+^ channels caused by the lipids SM-102 (1-octylnonyl 8-[(2-hydroxyethyl)[6-oxo-6-(undecyloxy) hexyl]amino]-octanoate), or by the addition of photo-sensitizers such as verteporfin or Di-8-ANEPPS (4-{2-[6-(dibutylamino)-2-naphthalenyl]-ethenyl}-1-(3-sulfopropyl)pyridinium inner salt) [156,157,158].

## 7. Conclusions

Ion channels continue to be of interest as targets for drug development, evaluating drug safety and diagnosis. Rhodopsin-based ion channels, in particular, are widely studied membrane proteins due to the ability to control their activity externally using light. To simplify the inherent complexity of the cellular environment, in vitro studies are used in which membrane proteins are isolated and reconstituted into model lipid membranes. This enables researchers to tune the overall composition of the lipid membranes one component at a time, and also to understand how individual membrane proteins are affected by their surrounding lipid environment. Measuring the passage of ions and electrical signals across the membrane of a lipid bilayer confirms that an ion channel is functional and can be done via both labelled and label-free assays. Ion responsive fluorescent dyes are the most commonly used assay, indicating changes in ionic concentrations, including protons, in real time under changing illumination conditions. Light-activated ion channels have been integrated to make functional complex synthetic systems such as artificial photosynthetic cells and synthetic tissues [116,159]. Further integration in the future of discoveries from in vitro light-controlled systems with those from in vivo studies will deliver insight into complex cellular events and also extend the frontiers of synthetic and chemical biology.

## Figures and Tables

**Figure 1 membranes-11-00857-f001:**
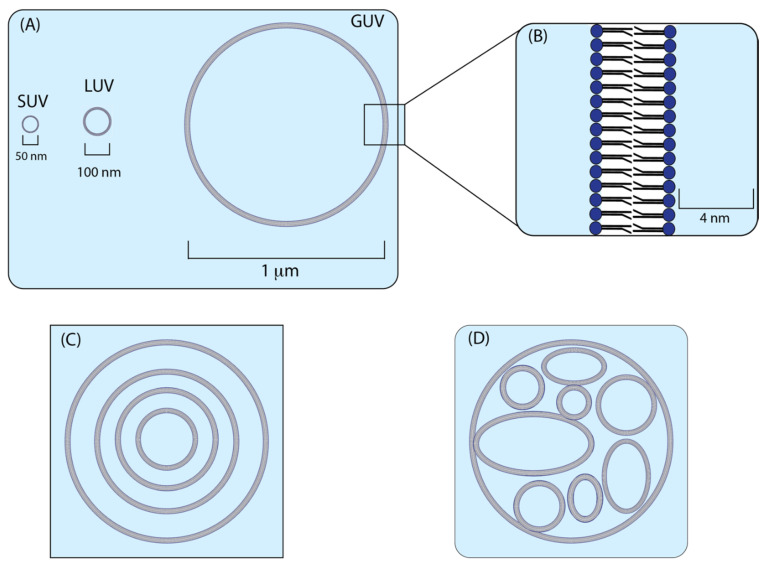
Schematic image demonstrating the structures of different liposome types. (**A**) Relative sizes and structures for the three types of unilamellar liposome vesicles. This includes small unilamellar vesicles (SUVs) (>100 nm), large unilamellar vesicles (LUVs) (100–500 nm), and giant unilamellar vesicles (GUVs) (~1 µm). (**B**) Membrane bilayer with the hydrophilic lipid head groups (dark blue) facing the internal and external aqueous environment (light blue). The inwardly facing hydrophobic tail groups (black) create a hydrophobic environment (white) within the membrane bilayer. (**C**) Structure of an MLV (multi-lamellar vesicle) including concentric internal bilayers. (**D**) structure of an MVL (multi-vesicular liposome) including non-concentric internal bilayers.

**Figure 2 membranes-11-00857-f002:**
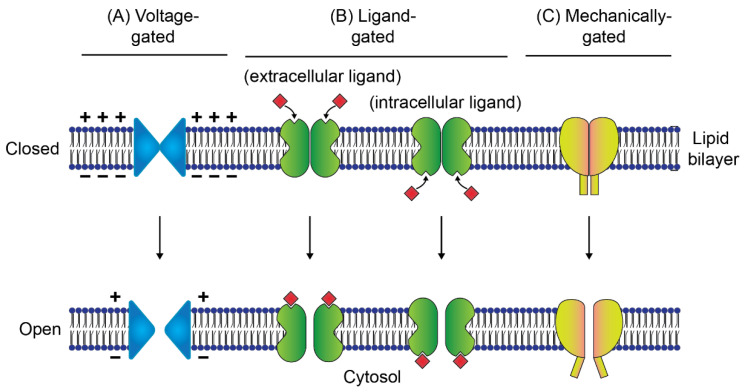
Types of gated ion channels. (**A**) Voltage-gated channels are ion-selective and open upon perturbation of the potential of cell membrane, (**B**) ligand-gated channels are sensitive to specific ligands and they are classified as extracellular and intracellular ligand-gated channels based on the position of ligands in the channel, and (**C**) mechanically gated channels sense forces and transduce the forces into electrical and/or chemical intracellular signals through the cell membrane and cytoskeleton.

**Figure 3 membranes-11-00857-f003:**
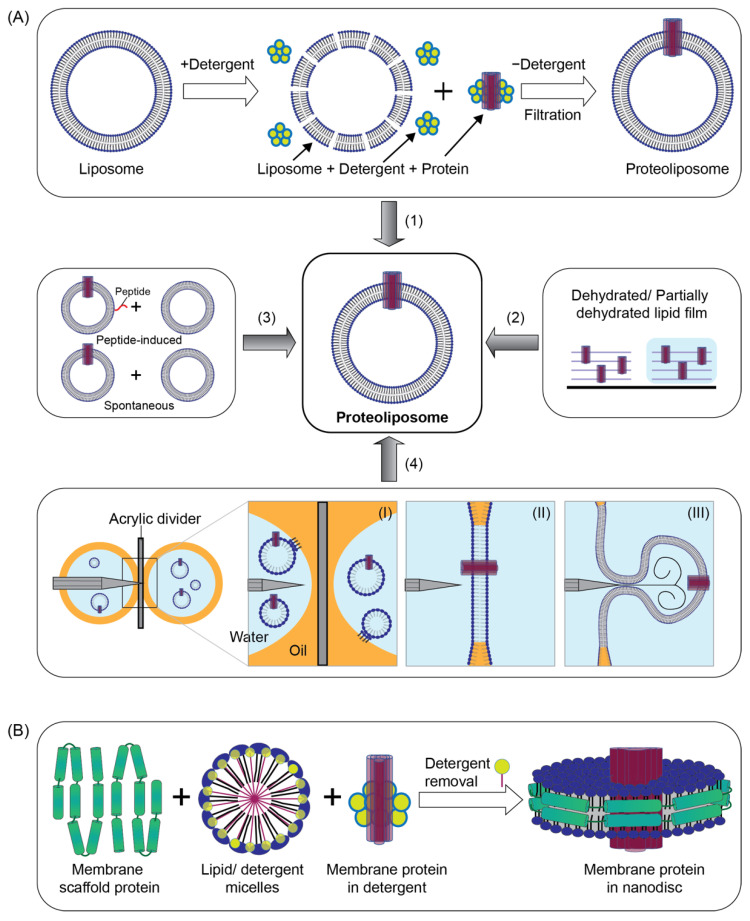
Schematic showing methods for reconstitution of membrane protein into model lipid bilayers. (**A**) Selected methods for proteoliposome preparation: (1) Direct reconstitution method: Preformed liposome and detergent solubilized proteins are mixed. Detergent facilitates the insertion of proteins in the lipid bilayers by a modest destabilization of the lipid vesicles. Finally, the detergent is removed by dilution, dialysis, or filtration, resulting in the formation of stable liposome formation with incorporated membrane proteins. (2) Dehydration–rehydration: Dehydrated lipid and protein solution are rehydrated either by spontaneous swelling or by an electric field. (3) Induced fusion: Two liposomes, usually one with already-reconstituted protein, are fused by peptide. (4) Microfluidic jetting: Two aqueous droplets are separated by an acrylic spacer in a chamber. A planar lipid bilayer with reconstituted protein is formed by merging the aqueous droplets when the spacer is removed. Proteoliposomes are generated from the lipid bilayer using the nozzle of an ink-jet printer. (**B**) Membrane protein reconstitution in nanodisc: Protein is reconstituted in a small patch of lipid bilayer surrounded by membrane scaffold protein.

**Figure 4 membranes-11-00857-f004:**
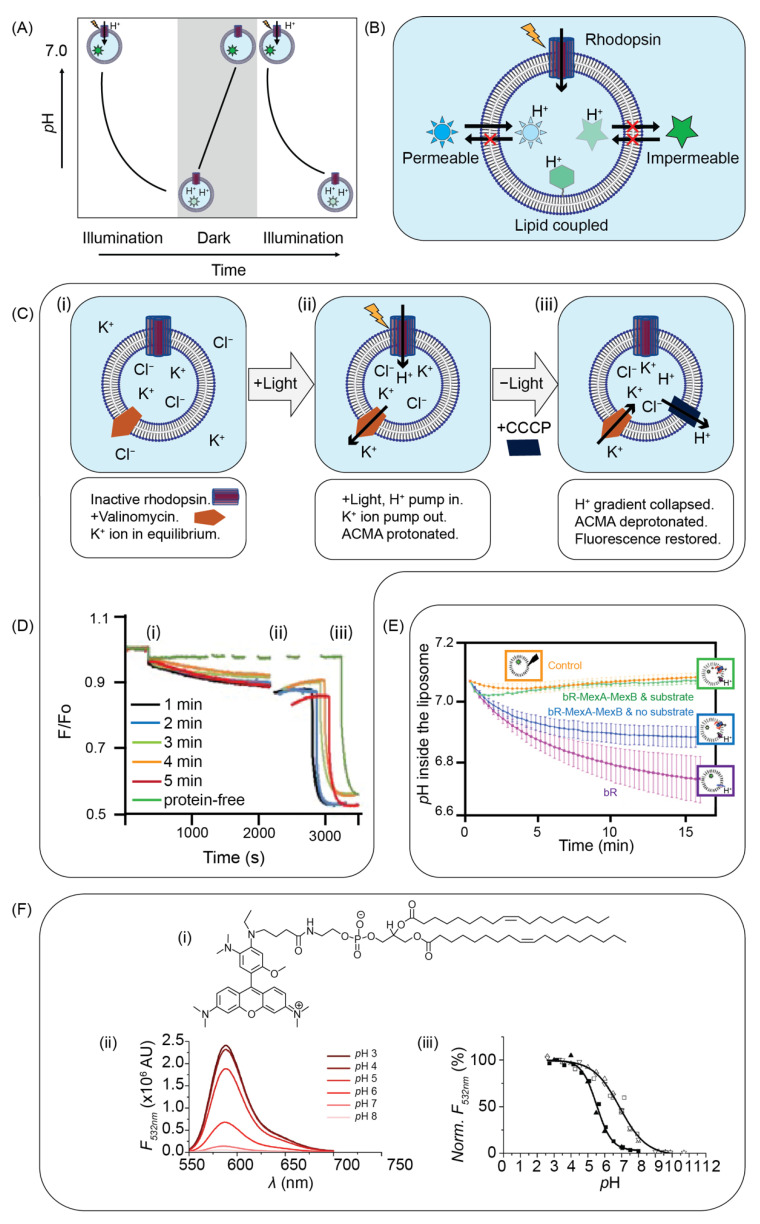
Fluorescence-based assays for characterizing ion channels. (**A**) Light–dark cycle of the photoactivity of rhodopsin—rhodopsin pumps protons as a result of light activation. In the dark, protons equilibrate by passive diffusion inside and outside of the liposome. Upon illumination again, rhodopsin activates, and proton flow recommences from outside of the proteoliposome. (**B**) Types of *p*H-sensitive dyes used to monitor proton flow: (1) hydrophobic membrane-permeable fluorescent dye, (2) water soluble membrane impermeable dyes, and (3) lipids with a *p*H-sensitive head group. (**C**) Schematic of the ACMA-based rhodopsin photoactivity assay: (i) in the proteoliposome mixture, ACMA, high concentration KCl, and valinomycin are added; (ii) upon illumination, rhodopsin-mediated proton pumping occurs pumping protons into the liposome which causes protonation and quenching of ACMA; efflux of K^+^ compensates for the charge imbalance arising from proton flow into the liposome; and (iii) finally, the addition of CCCP allows the free diffusion of protons across the membrane and thus the fluorescence intensity of ACMA drops rapidly. (**D**) Time-lapse recording of the fluorescence intensity relative to the initial fluorescence (F_O_), where “min” in the legend corresponds to the time in minutes. The numbering (i), (ii), and (iii) corresponds to the events described in Figure 4C. Reprinted with permission from ref. [111]. Copyright 2018 Elsevier. (**E**) Pyranine fluorescence assay: the MexAB protein transporter and bR were reconstituted into liposomes and *p*H change was monitored as function of time inside the proteoliposomes containing no bR (orange circles); bR (purple squares); and bR, MexB, and MexA without substrate (Hoechst 33342) (blue diamonds), and with substrate (green triangles) in the membrane. This subfigure is adapted from reference [125]. (**F**) *p*H-dependence of *p*Hrodo fluorescence: (i) chemical structure of *p*Hrodo-DOPE, (ii) fluorescence emission spectra of vesicle-embedded *p*Hrodo-DOPE in buffer solutions of different *p*H upon excitation at 532 nm, and (iii) *p*H-calibration curve of vesicle-embedded *p*Hrodo-DOPE (solid symbols) and water-soluble *p*Hrodo (open symbols). Adapted with permission from ref. [139]. Copyright 2015 Royal Society of Chemistry.

**Table 1 membranes-11-00857-t001:** List of *p*H-sensitive dyes commonly used to monitor proton pumping assays.

Dye	Type	λ_ex_/_em_, nm	Used to Monitor	References
ACMA	Membrane-impermeable, hydrophobic	410/490	bR	[116]
			ChIEF	[111]
			Hv1	[112]
			V-ATPases	[113,114]
Pyranine	Membrane-permeable, hydrophilic	454/520	bR	[83,86,125,126,127,128]
			pR	[134]
			sR	[129,130]
			xR	[131]
			ChR	[132]
			aR	[133]
			Xanthorhodopsin	[135]
			SbmA	[137]
Oregon Green 488-DHPE	Lipid-coupled	508/534	ATPase, Hv1	[142,143]
Fluorescein-PE	Lipid-coupled	498/517	Cytochrome c oxidase	[140]
*p*Hrodo-DOPE	Lipid-coupled	532/585	ATPase	[139,141]
Acridine orange	Hydrophobic	490/520	Synaptosome, acidic vesicular organelles	[144,145]
pHluorin	GFP variant	470/512	Cytosol and endoplasmic reticulum	[146]

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
