# Peer review of "Fluorescence Approaches for Characterizing Ion Channels in Synthetic Bilayers"

_membranes, 2021, doi:10.3390/membranes11110857_

Round 1
Reviewer 1 Report
The manuscript from Islam et al., reviews the fluorescence approaches to study ion channel activity in synthetic bilayers. Although this review is quite basic and does not make an extensive and deep collection of assays or methods it could be interesting for its publication if authors improve it.
My suggestions are the following:
- Nanodiscs are called at the manuscript and due to their importance nowadays it is mandatory to develop a section or subsection for them, as a reconstitution method and also to study membrane proteins and ions channels in particular at least structurally.
- Amphipathic polymers have become a very interesting alternative to detergents to solubilize membranes and reconstitute membrane proteins in nanodiscs. A section or subsections should be included reviewing their use.
- Cell-free systems are also a novel method to reconstitute membrane proteins that should be included in the review. It is faster and avoids the use of detergents.
- Section 5.2 is quite poor. It should be renamed as “Non-fluorescent assays for characterizing rhodopsins” since it is restricted to these proteins. The golden standard to characterize ion channel activity is the patch-clamp technique.
- The review is focused on rhodopsins but there is no figure describing their structure and function.
Other minor concerns are:
- Line 59, substitute “uses” by “used”
- Line 65, substitute “Figure 2B” by “Figure 1B”
- Section 4.1.1. The dehydration-rehydration method is performed to make giant liposomes from proteins reconstituted previously into liposomes. This has to be clarified in the text.
- Figure 3 footnote. The sedimentation method is included in the figure but it is not described in the text.
- Line 230, substitute “select” by “selected”
- Line 307, Rb+
- Figure 4C ii) H+; K+ ion pump out not in
- Figure 4D. The legend is not explained, what does min mean? Numbers in x-axes are not seen properly and are not leveled. Moreover, in the text, it is said that the addition of CCCP restores the fluorescence but it is dropped.
- Figure 4E. Does mexA-mexB use the H gradient to activate? and then the H gradient disappears?
- K+ at the footnote of Figure 4.
How is SPR applied to study rhodopsin? Though the binding of a ligand?? You should explain this point
Author Response
We thank the reviewer for careful reviewing. We have revised the manuscript according to your suggestions and feel that these helped to improve the quality of the manuscript. Here is an outline of our revisions.
Comment: Nanodiscs are called at the manuscript and due to their importance nowadays it is mandatory to develop a section or subsection for them, as a reconstitution method and also to study membrane proteins and ions channels in particular at least structurally.
Answer: Thank you for your suggestion. We have added a subsection as “4.2 Integrating membrane proteins into nanodiscs” in the section “4. Study of Ion Channels in Synthetic Liposomes” (page 6).
Comment: Amphipathic polymers have become a very interesting alternative to detergents to solubilize membranes and reconstitute membrane proteins in nanodiscs. A section or subsections should be included reviewing their use.
Answer: We thanks the reviewer for bringing this to our attention. We have added a subsection as “4.3 Integrating membrane proteins with detergent alternatives” in the section “4. Study of Ion Channels in Synthetic Liposomes” (page 7).
Comment: Cell-free systems are also a novel method to reconstitute membrane proteins that should be included in the review. It is faster and avoids the use of detergents.
Answer: We have added a subsection as “4.3 Integrating membrane proteins in cell-free systems.” in the section “4. Study of Ion Channels in Synthetic Liposomes” (page 7).
Comment: Section 5.2 is quite poor. It should be renamed as “Non-fluorescent assays for characterizing rhodopsins” since it is restricted to these proteins. The golden standard to characterize ion channel activity is the patch-clamp technique.
Answer: We agree with the reviewer that a discussion on patch clamping and electrophysiology is appropriate, especially as context for the limitations of fluorescence approaches. We have now renamed section ion 5.2 as “Non-fluorescent assays for characterizing rhodopsins” and have added a new section “6. Limitation of fluorescent assays for characterizing ion channels”. In this we directly emphasise that electrophysiology methods are more suited to characterising ion channel activity with regard to higher temporal and current sensitivity, but are typically lower throughput.
Comment: The review is focused on rhodopsins but there is no figure describing their structure and function.
Answer: Thank you for your comment. We have considered adding a panel to Fig. 4 showing the structure or cycle of rhodopsin. However, on reflection we realised that this topic has been covered in extreme detail elsewhere. For example, we now added a specific direction to existing review literature and a citation to Ernst et al. Chem. Rev. 2014, which is a 28 figure canonical review of this topic and direct the reader here. We could include eg. Fig 13 or Fig. 14 from that paper (below), and have received permissions to do so, or redraw from structures of other rhodopsins ourselves, but our feeling is this is not critical to our review here which is primarily about lipid and microscopy methods, and is very thoroughly described elsewhere. We feel it is not required here, but we are happy to add this as a subpanel on our Fig. 4 if the reviewer still prefer this to be included.
Other minor concerns:
- Line 59, substitute “uses” by “used”:
- Corrected (page 2)
- Line 65, substitute “Figure 2B” by “Figure 1B”: Corrected (page 2)
- Section 4.1.1. The dehydration-rehydration method is performed to make giant liposomes from proteins reconstituted previously into liposomes.
- This has to be clarified in the text: a sentence has been added in the text to clarify the point (page 6).
- Figure 3 footnote. The sedimentation method is included in the figure, but it is not described in the text:
- The figure 3 has been redrawn, where only commonly used reconstitution methods have been included (page 8).
- Line 230, substitute “select” by “selected”:
- Corrected (page 8)
- Line 307, Rb+:
- Corrected (page 10)
- Figure 4C ii) H+; K+ ion pump out not in:
- Corrected (page 11)
- Figure 4D. The legend is not explained, what does min mean? Numbers in x-axes are not seen properly and are not leveled. Moreover, in the text, it is said that the addition of CCCP restores the fluorescence, but it is dropped:
- The meaning of “min” in legend has been added and the text for CCCP has been clarified (page 12).
- Figure 4E. Does mexA-mexB use the H gradient to activate? and then the H gradient disappears:
- Yes, this is correct, mexA-mexB requires proton gradient to activate transport of the substrate, and this transport dissipates the proton gradient over time. We have expanded our explanation of this in the Figure legend.
- K+ at the footnote of Figure 4:
- Corrected (page 12)
- How is SPR applied to study rhodopsin? Though the binding of a ligand?? You should explain this point:
- We have added text to explain this further (page 13)

Reviewer 2 Report
Please see attached

Author Response
We thank the reviewer for their careful comments and have revised and improved our manuscript accordingly. Below we detail responses to specific comments.
Comment 1: Frankly speaking, the fluorescence approaches for investigations on trans-membrane ion channels, particularly on voltage-gated ion channels, in synthetic bilayers or liposomes could currently be limited to some extent (e.g., on the basis of time resolution). For example, the gating (activation, inactivation and deactivation) kinetics of voltage-gated sodium or sodium channels could for the time being be difficult to be examined in detail by fluorescence measurements. The issues relevant to limitations in this part of research are of particular importance, and hence they should be necessarily described in the paper. In essence, the time scales used for fluorescence measurements (e.g., gating kinetics) could be overly distinguishable from those for prototypical electrophysiological ones (i.e., voltage-clamp studies).
Answer: We thank the reviewer for this comment. Indeed, it is worth mentioning the limitations of fluorescence approaches and contrasting and comparing them with the 'gold-standard' that is electrophysiology measurements. The most notable limitations of fluorescence approaches are the lack of temporal resolution in comparison with patch-clamp type methods, as well as the fact that the methods do not report direct movements of ion or currents, as electrophysiology methods do. As a result, fine changes in the current flow, such as sub-steps in channel activation cannot be resolved. We completely agree this was lacking from the previous manuscript and have added now a section “6. Limitation of fluorescent assays for characterizing ion channels” to discuss this (page 14).
Comment 2: Please state some possible dyes, if any, tailored for the measurements in conformational changes of voltage-gated sodium or calcium channels in the manuscript, though cytosolic Ca2+ indicators (e.g., Fluor-3) could be commonly used. Is it possible to identify the open, activation, and inactivation conformational states in voltage-gated ion channels by virtue of fluorescence approaches?
Answer: The conformational states that ion channels adopt during activation and inactivation are major target for the characterization techniques. As stated above, the reviewer raises a godo point and activation and inactivation sub-steps cannot typically be resolved by the fluorescence-based assays. This limitation has been discussed in the newly added section “6. Limitation of fluorescent assays for characterizing ion channels” (page 14).
Comment 3: In line 211, please include an additional sentence along with a reference. That is, “Of note, the ingredients (e.g., SM-102 [(1-octylnonyl 8-[(2-hydroxyethyl)[6-oxo-6-(undecyloxy)hexyl]amino]-octanoate)]) for the formation of lipid nanoparticles may potentially modify the magnitude of a unique population of K+ channels (Cho et al., Biomedicines 2021;9(10):1367).” It also needs to be noticed that DOPE or DHPE per se might also directly influence the activity or kinetics of varying types of membrane ion channels, particularly at voltage-gated ionic channels. Such perturbations on the magnitude and/or kinetics of ion channels might not be detected simply by fluorescence approaches. This issue regarding the limitations is virtually important, and it hence needs to be described and included in the revised manuscript somewhere.
Comment 4: Verteporfin, a photosensitizer, has been previously reported to modify membrane ionic currents (Huang et al., Front Chem 2019;7:566). The paper needs to be incorporated as well.
Comment 6: Please also add an reference (effect di-8-ANEPPS on BKCa channels) into the manuscript somewhere (Pflugers Arch 2008;455:687-699).
Answer: It is worth mentioning that some important subtleties of ion channels cannot be detected using fluorescent methods, we agree. We have included all 3 citations now in the newly added section “6. Limitation of fluorescent assays for characterizing ion channels” (page 14).
Comment 5: Is it possible that SUVs, LUVs or GUVs can be examined by standard patch-clamp recordings? Please elaborate the advantage and disadvantage of electrophysiological measurements on liposomes, as compared with fluorescence approaches.
Answer: As above, we have now added a section on advantages and disadvantages of electrophysiological measurements on liposomes in the newly added section “6. Limitation of fluorescent assays for characterizing ion channels” (page 14). Indeed, GUVs and SUVs can be patch clamped, but this is limited only to vesicles of a certain size, typically GUVs, due among other reasons to the size of the patch but also the need to find the liposome using bright field microscopy.
Comment 7: In Page 12, valinomycin appearing in Figure 4C needs to be corrected.
Answer: Corrected.
Round 2
Reviewer 1 Report
Authors have answered properly to my suggestions.
Reviewer 2 Report
The investigators have answered most of the comments raised.